# Explaining Redundancy in CDK-Mediated Control of the Cell Cycle: Unifying the Continuum and Quantitative Models

**DOI:** 10.3390/cells11132019

**Published:** 2022-06-24

**Authors:** Daniel Fisher, Liliana Krasinska

**Affiliations:** Institut de Génétique Moléculaire de Montpellier (IGMM), Centre Nationale de la Recherche Scientifique (CNRS), Institut National de la Santé et de la Recherche Médicale (INSERM), University of Montpellier, 34293 Montpellier, France

**Keywords:** cyclin-dependent kinase, redundancy, specificity, cell cycle control

## Abstract

In eukaryotes, cyclin-dependent kinases (CDKs) are required for the onset of DNA replication and mitosis, and distinct CDK–cyclin complexes are activated sequentially throughout the cell cycle. It is widely thought that specific complexes are required to traverse a point of commitment to the cell cycle in G1, and to promote S-phase and mitosis, respectively. Thus, according to a popular model that has dominated the field for decades, the inherent specificity of distinct CDK–cyclin complexes for different substrates at each phase of the cell cycle generates the correct order and timing of events. However, the results from the knockouts of genes encoding cyclins and CDKs do not support this model. An alternative “quantitative” model, validated by much recent work, suggests that it is the overall level of CDK activity (with the opposing input of phosphatases) that determines the timing and order of S-phase and mitosis. We take this model further by suggesting that the subdivision of the cell cycle into discrete phases (G0, G1, S, G2, and M) is outdated and problematic. Instead, we revive the “continuum” model of the cell cycle and propose that a combination with the quantitative model better defines a conceptual framework for understanding cell cycle control.

## 1. How Many Separate Phases Are There in the Cell Cycle?

There is an old joke, with many variants, that goes along these lines: how many (administrators/politicians/cell biologists) does it take to change a light bulb? (The reader can find the answer at the end of this article*). In this review, we will attempt to answer the question of how many cyclin-dependent kinases (CDKs) (and cyclins) does it take to run the eukaryotic cell cycle? The popular molecular biology model of regulation of the mammalian cell cycle invokes specific CDK–cyclin complexes to sequentially trigger separate transitions (G0-G1, G1-S, S-G2, G2-M) (Figure 1; for more detail, see [1]), although this is arguably based on an accumulation of the correlation and circumstantial evidence rather than a clear demonstration of causality. However, before describing these different CDK–cyclin complexes and the models of CDK-mediated cell cycle regulation, it is worth revisiting the evidence that such cell cycle transitions actually exist.

First, however, we should define what we understand by “the cell cycle”. Do we mean a budding yeast cell cycle, in which the mitotic spindle forms in G1 and there is no clear G2 phase? A fission yeast cell cycle, in which a distinct G1 phase is undetectable during exponential growth? A cell cycle in early frog embryogenesis, in which there is no cell growth but just a succession of DNA synthesis and chromosome segregation phases? Or a mammalian somatic cell cycle? If it is the latter, do we mean a cell cycle of murine embryonic fibroblasts or embryonic stem cells, human cervical cancer cells, or cells dividing in a living animal? Does it matter anyway: will these different cell cycles rely on different CDK or cyclin complexes? Will the answer also be different between cells that have limited or abundant growth resources?

Our current thinking about the cell cycle has been heavily influenced by a few key studies, on which current models have been built. The “single cell analysis” by Howard and Pelc in dividing cells of the faba bean shoot showed that DNA was synthesized in a limited part of the cell cycle [2], which they later named as the S-phase, thus dividing the cell cycle into four phases: G1, S-phase, G2, and mitosis. Another phase, G0, was added shortly thereafter to describe the non-proliferating cells, which were assumed to be in some biochemically distinct form of a quiescent state. This emerged from experiments of tritiated thymidine incorporation in mouse intestinal epithelium [3], mice bearing spontaneous mammary tumors [4], regenerating rat liver [5], and mouse stomach epithelium [6], which demonstrated the existence of cells dividing either very slowly or not at all. Such apparent quiescence could also be induced experimentally in the cell culture of non-transformed (but not cancer) cells by contact or density inhibition [7,8,9], deprivation of nutrients [10] or serum [11], or the elevation of cyclic AMP levels [12]. Treating cells in these different ways prevented DNA synthesis, thus assigning the arrested cells to G1, while after restoring the growth conditions, DNA synthesis was initiated after a similar delay; however, the cells could not restart if transferred into an alternative condition that did not support growth [13]. The conclusion of these experiments was that cells subjected to various “physiological” deprivations exited the cell cycle post-mitotically and prior to a “restriction point” that became synonymous with the G0–G1 transition, and, more recently, with a “decision point”, whose passage depends on the level of CDK activity [14].

However, whether the restriction point exists, and whether the G0 phase is a separate biochemical state, or simply a very long form of G1, has long been debated [15,16]. Early experiments are consistent with the latter, since long autoradiographic exposure failed to reveal cells that did not incorporate nucleotide precursors, and demonstrated that G1 could last considerable lengths of time: up to 90% of the 140 h cell cycle of hamster cheek epithelium [17]. Through time-lapse microscopy, Killander and Zetterberg [18] noticed that the length of G1 was highly variable, explaining the majority of the variation in doubling time (which varied from 12 h to 28 h). Importantly, there was smaller variability in the size of cells initiating DNA synthesis, suggesting that a parameter related to growth is necessary for cells to replicate DNA. After serum stimulation of the arrested hamster fibroblasts, while there was a delay before the first round of DNA synthesis (thus, a G1 phase), there was no such detectable G1 phase in the subsequent cell cycle [19]. These seminal experiments argue that G1 is not an integral part of the cell cycle, but a facultative period during which the cell acquires resources for DNA synthesis, and whose length, and even existence, depends on previous environmental conditions. As such, the restriction point should not exist. Indeed, a simple explanation for the fact that release from physiological arrests led to a delay before onset of DNA replication, and which is not seen during exponential growth, is that growth rates (macromolecule biosynthesis rates) take time to get back up to levels that allow for the rapid firing of replication origins. Thus, there is no need to invoke a transition, or discrete event, during G1. This view of G1 control has been called the “continuum” model by Stephen Cooper, according to which G1 is simply a continuation of growth conditions in the rest of the cell cycle without any unique biochemical transitions [20]. A key aspect of this model, which has received insufficient attention, is the principle of cell age order invariance, in which any treatment (such as serum starvation) of a population of cells will affect all cells equally and not just those in the early G1 phase. While the first division of the latter may be substantially affected, the rest of the cells divide on schedule since the growth parameter is only limiting to initiate DNA replication. However, in the second cell cycle, the altered growth, which occurs in all phases of the cell cycle “catches up” with cells that had previously already engaged in DNA replication at the time of treatment, and the order of cell division was maintained. An important implication of this reasoning is that it is impossible to synchronize cells by population treatments, since the age order cannot be altered.

Whether the S-phase and G2 are separate, discrete phases of the cell cycle was also questioned early on, but these experiments have been disregarded over time. Shackney and Ford demonstrated a gradual rise and fall in the DNA synthesis rates during the interphase and argued that the onset and termination of DNA replication are not discrete events [21,22]. This is consistent with more recent experiments that make use of molecular genetics. Although slowing down DNA replication in budding yeast delays mitotic onset by activating the S-phase checkpoint [23], the presence of unreplicated chromosomes per se is not sensed and does not delay mitosis [24], while DNA replication can occur, even in late mitosis [25]. These findings arguably remove the need to invoke G2 as a separate phase of the cell cycle (i.e., after replication has been completed and in preparation for mitosis). Smith and Martin [26] even questioned whether cells cycle at all, in the sense that they undergo a cyclic sequence of events, or whether they stochastically flip from one state (G1) to another (the sequence of S, G2, and M).

The variability of time between cell divisions may not be solely due to a different length of time prior to DNA synthesis. In lymphocytes, the duration of all phases varies considerably and is proportional to the inter-division time, with the majority of cell cycle time variability resulting from the combined length of the S/G2/M phases [27]. In this regard, with the exception of mitosis, cell cycle phases are not distinct phases. This suggests a simple continuum model, whereby the growth conditions promote the activity of an overall regulator such as CDK activity, determining the rate of progression through DNA synthesis, the only abrupt transitions being the entry into, and exit from, mitosis (Figure 2).

## 2. Building the Multiple CDK Model of Cell Cycle Control

The history of the discovery of cell cycle control by CDKs has been recently reviewed [1]. Briefly, in early work, Masui and Markert identified a self-amplifying biochemical component of *Xenopus* eggs, “metaphase promoting factor, MPF”, which triggered the onset of meiotic metaphase when injected into recipient arrested eggs [28]. Independently, Hartwell and colleagues established the dependency relationships of budding yeast cell division cycle (CDC) mutants that they had been isolating, and identified one mutant (Cdc28) arrested prior to DNA replication, leading to the description of this point in the cell cycle by the term “Start” [29], analogous to the idea of the restriction point in mammalian cells. In the same year, E. Morton Bradbury and colleagues discovered that the phosphorylation of histone H1 (then called F1) rises dramatically at mitosis in the naturally synchronous nuclear cycles of the slime mold *Physarum polycephalum* [30], leading to the prescient hypothesis that the enzyme responsible (later found to be CDK1-cyclin B) drives mitotic onset [31]. “Cyclins” (proteins periodically translated and degraded during cell cycle) were first seen in experiments on the control of mRNA translation in sea urchin and clam eggs, and later defined cyclins A and B; injection of the mRNA into *Xenopus* oocytes promoted the acquisition of MPF [32]. Subsequent molecular cloning developments have identified the genes encoding Cdc28 and the fission yeast homologue Cdc2 as protein kinase homologues [33,34]. The realization that the mitotic histone kinase contains a homologue of Cdc2 [35,36] and cyclin [37], and the final demonstration that the mitotic promoting factor is a heterodimer of cyclin and Cdc2 [38], defined the first CDK. Up to this point, the picture seemed fairly straightforward, with CDK1–cyclin controlling the cell cycle. However, a flurry of research into CDKs and their regulation over the subsequent decade painted a much more complex scene. A second Cdc2 homologue was soon identified in *Xenopus* by the screening of mRNAs specifically enriched in eggs [39]. Quickly after, a whole Cdc2 multi-gene family, renamed CDKs, was identified in human cells [40], and several classes of new metazoan cyclins were identified (notably, D and E-type), by complementation of budding yeast mutants for genes encoding the three Cln cyclins [41,42,43]. The second metazoan Cdc2 homologue, renamed CDK2, was originally reported to promote the onset of S-phase, largely based on data from the *Xenopus* egg extracts [44], followed by work using the microinjection of anti-CDK2 antibodies [45] or dominant negative mutants [46] in mammalian cells. The latter paper also proposed that a third, vertebrate-specific close relative of CDK2, CDK3, promotes progression through G1, while another study concluded that CDK3 promotes G1-dependent transcription by E2F in cells with high c-Myc levels [47]. CDK3 functions have largely been ignored, in part because in laboratory strains of mice, it is inactivated by truncating mutation [48], demonstrating that it is dispensable for animal development and therefore the cell cycle.

It is now apparent that there are 26 CDK1-related kinases and 30 genes with significant homology to cyclins in the human genome (HUGO) [49], but not all CDKs are activated by a cyclin subunit, nor do all cyclin subunits have cyclic expression in the cell cycle. Indeed, most CDKs and cyclins are not involved in cell cycle control, and have many other functions in fundamental cellular processes [50,51,52]. We will hence restrict our considerations of CDKs and cyclins to those (CDK1,2,3,4,6 and cyclins A, B, D, and E) generally considered as cell cycle regulators.

## 3. “G1” Cyclin–CDK Complexes May Control Growth Rather Than S-Phase Onset

Early work on CDK4 (which is present in all metazoans) and CDK6 (which is specific to vertebrates) found that these kinases associate with D-type cyclins and phosphorylate the Retinoblastoma (RB) tumor suppressor protein [53,54,55]. Since hyperphosphorylated RB cannot repress E2F-dependent transcription, the obvious interpretation was that CDK4/6 inactivates RB and thus drives passage through the restriction point, which by now had become virtually synonymous with RB inactivation. The idea of CDK-mediated inactivation of RB controlling the restriction point and entry into the cell cycle was also extended to CDK3–cyclin C complexes. This conclusion was based on somewhat accelerated or delayed general RNA synthesis from CDK3 over- and under-expression, respectively, as well as from the interference with kinase-dead CDK3 [56]. It has not been confirmed by subsequent studies, and it is clear that CDK3 is not essential for progression into the S-phase in any cell line. In any case, there is little concrete evidence that there exists a G1 restriction point controlled by RB phosphorylation in cycling cells. For example, the timing of RB phosphorylation and the placement of the putative restriction point did not coincide [57], although these experiments compared the serum starved and released cells with exponentially growing cells. Furthermore, in cells released from growth arrest, RB is not inactivated by CDK4/6-mediated phosphorylation, but by hyperphosphorylation, which, upon the stimulation of growth-arrested cells, occurs coincidently with the onset of the S-phase and active CDK2 complexes [58,59]—the opposite conclusion to earlier work [60]. In cycling cells, RB phosphorylation and inactivation are maintained throughout the cell cycle, and long-term cell cycle arrest triggered by RB dephosphorylation can occur in G2 [61]. Analogously, new research suggests that even in budding yeast, acute nutrient deprivation can provoke growth arrest independently of the cell cycle stage [62].

If, as suggested [15,16], the restriction point does not exist, and G1 is not a discrete phase of the cell cycle, but a variable period before exponential DNA replication origin activation occurs, then the so-called G1 cyclins might not be required for the cell cycle. There is strong evidence that this is indeed so. In *Drosophila*, which possess single genes encoding a homologue of CDK4/6 and of cyclin D, these genes are required for the accumulation of cell mass but not passage through the cell cycle [63,64], and the complex promotes mitochondrial protein synthesis [65]. Mice lacking CDK6 are viable, while double knockouts of CDK4 and CDK6 allow for much of the embryonic development, and double mutant fibroblasts had no differences in the cell cycle kinetics even after mitogen withdrawal and re-stimulation [66]. Similar results were obtained in mice lacking cyclins D1, D2, and D3 [67]. While the normal proliferation of fibroblasts lacking D-type cyclins depends on cyclin E1 or cyclin E2, proliferation of most other cell types could occur in the absence of all D- and E-type cyclins [68]. Thus, these data suggest that there is no universal requirement for so-called G1 cyclins in mammalian cell cycle control.

Still, D- and E-type cyclin overexpression shortens G1 [69,70]. One explanation could be that they trigger the early inactivation of RB, thus accelerating the accumulation of overall CDK activity and DNA replication origin firing. If so, then cells should have a shorter cell cycle, but this was not observed; in contrast, there was evidence for an increase in the length of the S-phase. This may be due to reduced licensing of replication origins (assembly of MCM complexes) due to excessive CDK activity, leading to fewer replication forks and therefore a lengthened S-phase, during which growth also occurs. As a consequence, G1 is shortened, since growth conditions were attained in the longer S-phase of the previous cell cycle. An alternative explanation that is compatible with these observations is that overexpressed D- and E-type cyclins promote growth throughout the cell cycle, removing the delay before macroscopic DNA replication is observed. This is consistent with cyclin D synthesis, correlating with mitogen signaling throughout the cell cycle with knock-on effects in the subsequent cell cycle [71]. Regardless of the mechanism of the effects of cyclin overexpression, it is nevertheless clear that so-called G1 cyclins and CDKs are not essential for cell cycle progression in most cells.

## 4. Specific Cyclins and CDKs Modulate Kinetics of Overall CDK Activity

The inter-relationships between other cell cycle CDKs and cyclins and their regulators are understood in kinetic detail (see, for example, [72,73]). The organization of this network, which has been compared to a wiring diagram, is complex. It involves the temporal control of the transcription and proteolysis of specific cyclins and of small protein CDK inhibitors (CKI), auto-amplification loops with positive and double-negative feedback, and, above all, control of the protein phosphatases that reverse CDK-mediated phosphorylation. This complexity precludes intuitive understanding of the effects of the modulating activity of any one component; mathematical modeling revealed a generic network architecture in which coupled double-negative and three-component negative feedback loops underlie cell cycle transitions, critically, the system determines the overall balance between CDK activity and the activity of the opposing protein phosphatases [74].

In agreement with this idea, there is plenty of evidence, in systems that have very different cell cycle kinetics, that direct modulation of the overall CDK activity can bypass the requirements for much of the network. The most obvious demonstration comes from genetic simplification of cell cycle regulators in fission yeast. Early experiments, when S-phase-promoting cyclins had not yet been identified, provided evidence that the deletion of the mitotic cyclin B reset cells from G2 and promoted re-replication, while the same mitotic cyclin B in complex with CDK1 was sufficient to promote entry into the S-phase in the absence of other cyclins [75,76]. Given that the S-phase occurred when CDK activity was low and the M-phase when CDK activity was high, the simplest explanation is that the level of CDK activity determines the order and timing of cell cycle events, which became known as the “quantitative model” of cell cycle control. This was later proven by deleting all known “cell cycle” cyclins and CDK1, and replacing them with a monomolecular CDK-mitotic cyclin B fusion protein, which allowed for the generation of viable cells with no major differences in cell cycle kinetics [77]. Importantly, engineering sensitivity to a chemical ATP-analogue inhibitor into this CDK fusion complex, while deleting the destruction box causing mitotic cyclin degradation, allowed us to artificially alter the cell cycle position at will by washing in and out the different concentrations of the inhibitor: the artificial induction of low CDK activity triggered the S-phase and high activity triggered mitosis, even if the S-phase had not been completed. Thus, modulating the overall CDK activity can drive the cell cycle, independently of the regulatory network. This system was also used to show that cell cycle-linked periodic transcriptional oscillations, which contribute to the dynamics of CDK activation at different cell cycle phases, are not independent of the CDK oscillator, as originally proposed [78,79], but are intrinsically controlled by the overall CDK activity [80].

It remains unclear how the normal temporal separation of the S-phase and M-phase occurs. The key is the missing phosphatase activity. A theoretical consideration of futile cycles (systems that operate with enzymes that mutually oppose each other’s activity) shows that their net output is highly sensitive to small changes in the activity of any of the enzymes involved [81]. Applying this logic to the cell cycle suggests that mitosis does not normally happen when CDK activity promotes S-phase because mitotic substrates are dephosphorylated by interphase phosphatase activity [82] (Figure 3). There is now good evidence that this is the case. In replicating the *Xenopus* egg extracts that cannot enter mitosis because cycloheximide prevents cyclin translation, the simple inhibition of the phosphatase that reverses CDK-mediated phosphorylation was sufficient to trigger mitotic entry (while simultaneously inhibiting DNA replication), demonstrating that it is active in the S-phase [83]. Further manipulation of CDK1 activity (e.g., by adding cyclin B or by depleting CDK1) simply alters the timing of this transition. Quantitative phosphoproteomics in fission yeast also proves the existence of active phosphatases in the S-phase since specific CDK inhibition causes a rapid decline in substrate phosphorylation in vivo [84]. In mammalian cells, the principal phosphatase that reverses CDK-mediated phosphorylation was identified a long time ago as PP2A [85]. PP2A activity is bistable in cycling *Xenopus* egg extracts, increasing during the interphase in parallel with increasing CDK activity, and dropping drastically at mitotic entry [86] (Figure 3). This surge in PP2A activity boosts futile cycling in the interphase, making for a highly sensitive substrate phosphorylation switch, even when CDK activity changes more moderately. Thus, much of the CDK substrate phosphorylation at mitotic entry may be due to phosphatase inhibition rather than CDK-cyclin activation. It is not yet known how phosphatase activity is related to ongoing DNA replication, but it is likely that it is tied to the maintenance of a low CDK activity. The latter depends on CDK1 tyrosine-15 phosphorylation regulated by replication-mediated basal checkpoint kinase activities (reviewed in [87]).

## 5. Redundant CDK-Mediated Control of S-Phase Onset

Why are specific CDKs often thought to be required to trigger DNA replication? Although early work reported that CDK2 is essential for DNA replication in *Xenopus* egg extracts [44], later experiments employing single-molecule analysis of DNA replication origin firing showed that removing or inhibiting CDK2 only slows DNA replication (by reducing the frequency of replication origin firing), but does not abrogate it, as CDK1 also contributes [88,89]. Importantly, when CDK levels are low, the frequency of replication origin firing correlates with the CDK levels, implying that the CDK-mediated control of the S-phase is not “all or nothing”, at least in an embryonic system, but rather more quantitative and continuous [88]. Without manipulating the system, in egg extracts, CDK activity at the S-phase exceeded the requirements, ensuring the extremely rapid DNA synthesis that occurs in early embryonic cell cycles. Similarly, in mammalian cells, the need for CDK to bypass a restriction point in G1, usually equated with RB phosphorylation and inactivation, could also be explained by a RB-controlled positive feedback loop amplifying CDK activity early in the cell cycle to promote the efficient firing of replication origins. This might give the impression of an all-or-nothing switch for the S-phase onset, although careful single cell analysis in early experiments showed that S-phase onset is more continuous.

Assuming the conservation of these principles, this continuum model for the CDK control of DNA replication explains why mice lacking CDK2 are viable, which was, at the time, a very surprising result [90,91]. Indeed, follow up work showed that, in the absence of CDK2, CDK1 is required for the S-phase, and it binds to cyclin E, which is present in the early but not late part of the cell cycle [92]. Similar conclusions were derived from the chemical genetic analysis in chicken DT40 cells [93]. As expected, CDK1 was found to be essential for mitosis in mice, whereas simultaneous elimination of all interphase CDKs (CDK2, 4, and 6) had little effect on the cell cycle [94]. A recent paper directly demonstrates that CDK1 activity is also involved in promoting replication origin firing in mammalian cells, as previously seen in *Xenopus* [88]. Using a chemical genetic approach, Suski et al. [95] found that specific inhibition of CDK1 eliminated a subset of phosphorylations on the replicative helicase subunit MCM2, and reduced the number of replication origins in asynchronous ES cells. Furthermore, most mammalian cell types could still cycle, albeit more slowly, in the absence of CDC7, another conserved kinase previously thought to be essential for triggering the S-phase by phosphorylating MCM proteins. In these circumstances, CDK1 became essential for the onset of the S-phase, while CDK2 was dispensable. 

## 6. Specific CDK–Cyclin Complexes Are Not Essential for Entry into Mitosis

For a long time, there existed a potential problem with the application of the quantitative model to mammalian cells, in which CDK1 and CDK2 are at least partly interchangeable: in mice, CDK2 is incapable of inducing mitosis in the absence of CDK1 [96], even when expressed from the endogenous *CDK1* locus, in which case it is also unable to support correct germ-tissue development [97]. There are two possible explanations: either there exists qualitative differences between the two kinases and their control of S-phase and mitosis in mammalian cells, or, more likely, quantitative differences in the levels of kinase activity attained by CDK1 and CDK2. In the latter case, CDK1 can present the relatively low levels required for replication origin firing but CDK2 cannot reach the high levels required in mitosis. That the latter explanation is correct was recently demonstrated by elegant genetic studies in human non-transformed and cancer cells, in which single and double CDK1 and CDK2 degron mutants were superimposed onto a CRISPR-Cas9-mediated knockout background for each kinase, and CDK1, CDK2 or both were then acutely depleted [98]. This showed that CDK1 can substitute for CDK2 in triggering the S-phase, while, in cancer cells, CDK2–cyclin B complexes allow for the onset of mitosis in the absence of CDK1. However, cells could not complete mitosis, and the phosphorylation of CDK substrates was low, while the APC/C cyclin degradation machinery could not be turned on. All of these phenotypes including cell viability were rescued by over-expressing CDK2. Interestingly, in non-transformed RPE1 cells, CDK2 levels were lower, and did not suffice to trigger mitotic entry upon CDK1 degradation. Again, this could be restored by CDK2 overexpression. Thus, these results suggest that there are only quantitative differences in the ability of different CDKs to the control, S-phase, or mitosis. This principle is conserved in fission yeast, where the S-phase CDK–cyclin complex can trigger mitosis [99,100].

What about cyclins? In *Xenopus* embryonic cell cycles, cyclin A has been proposed to be a mitotic initiator [101], by virtue of its phosphorylation of Bora, which then activates the cascade of Aurora and polo-like kinases (Plk), the latter of which promotes the auto-amplification loop of CDK1–cyclin B by activating CDC25 [102]. In other words, adding complexity to the system (cyclin A, Aurora, Plk1) alters the kinetics of CDK1–cyclin B activation (overall CDK activity), but does not qualitatively change the system (e.g., by allowing phosphorylation of a key substrate for mitosis that CDK1–cyclin B cannot phosphorylate). Cyclin A can also trigger mitosis in the absence of cyclin B in this system [103]. In mammalian cells, cyclin A2 is required for the proliferation of some cell types (hematopoietic cells and embryonic stem cells) but not others (fibroblasts), in which cyclin E is expressed; the combined loss of both cyclins E and A is lethal [104]. Similarly, genetic inactivation of both E-type cyclins in mice does not result in cell cycle arrest [105], but, rather, hinders the restart of the S-phase after serum starvation, which might implicate a general role in promoting growth. These observations rule out the possibility that cyclin A or cyclin E have specific functions in the cell cycle that cannot be fulfilled by other complexes. The question of why cyclin B does not suffice to trigger the S-phase onset in the absence of cyclins A and E also does not appear to be due to the substrate specificity; simply, insufficient levels of cyclin B are present in the nucleus. In both *Xenopus* embryonic cell cycles [106] and human somatic cell cycles [107], appending a nuclear localization signal to cyclin B1 restores efficient DNA replication upon the loss of cyclins E or A, respectively. Furthermore, in the presence of CDK2, CDK1 associates only with cyclin B during early stages of the cell cycle, but not with cyclins E or A, and low levels of cyclin B1 are detected in the nucleus [95].

Is it possible that B-type cyclins are not required for the G2/M transition? Indeed, early work in *Drosophila* suggested that neither of the two B-type cyclins (B and B3) are essential for mitotic onset [108,109], while in their presence, cyclin A is not required [110], although cells could not complete mitosis in the absence of B-type cyclins. In mammalian cells, as with CDKs [98], acutely induced degradation of cyclin A2 protein in G2 phase via degron activation prevented mitotic entry, whereas acute loss of both B-type cyclins (B1 and B2) did not [107]. This requirement for cyclin A2 cannot be explained by the essential role of the Aurora/Plk pathway in the mitotic onset in this system, since Plk inhibition does not prevent entry into mitosis, but rather arrests cells within mitosis [111], consistent with the mutant phenotypes of *Polo* in *Drosophila* [112]. Mitotic entry in the absence of cyclin A2 in mammalian cells could be restored by increasing CDK1–cyclin B activity or reducing the activity of PP2A. In the absence of B-type cyclins, cyclin A2 suffices to inhibit PP2A via the activation of Greatwall kinase, which then phosphorylates and activates the PP2A inhibitors Arpp19/ENSA [107]. This again implies that the overall CDK/PP2A activity ratio is the rate-limiting factor for mitotic onset, and that there are no qualitative requirements for any single cyclin. However, the absence of B-type cyclins prevents mitotic progression (as in *Drosophila* cyclin B mutants), probably because cyclin A is degraded early in mitosis, leading to insufficient CDK activity to trigger the activation of the APC/C and to complete mitosis.

Although it is conceivable that genetic compensation (i.e., the overexpression of compensating proteins in genetic mutants [113]) could be responsible for the lack of lethal phenotypes of certain CDK or cyclin mutants in particular systems, this would necessarily mean that compensation by other proteins is possible, and therefore invoke functional redundancy. Nevertheless, similar results obtained by acute depletion using recent degron approaches rule out the argument that genetic compensation overrides requirements for CDK and cyclin specificity.

## 7. Conclusions

We have seen that the prevailing model of cell cycle control, in which sequential activation of different CDK–cyclin complexes promotes successive transitions between discontinuous cell cycle phases, fails to account for most of the observations resulting from the genetic analysis of CDKs and cyclins, irrespective of the organism studied or particular experimental approach used. If absolute substrate specificity is hard-wired into each complex, it would be hard to explain how one complex could simultaneously phosphorylate many highly divergent proteins to control diverse biological processes. Differences in the affinity of specific CDK–cyclin complexes for particular substrates certainly exist, and result in part from interactions between short linear motifs and identified amino acids on particular cyclins [114,115,116]. However, it is clear that this is not the principal determinant of the phosphorylation kinetics for many substrates. Mitotic CDK substrate phosphorylation occurs in a switch-like manner, indicating that CDK activity is largely in excess of the requirements for complete phosphorylation. Thus, the comparatively limited differences in the affinities for different substrates become unimportant for the overall kinetics. Nevertheless, different CDK–cyclin complexes become activated and inactivated with different mechanisms and timing. This determines the overall CDK/phosphatase activity ratio present at any one time, and also contributes to the irreversibility of progression through mitosis [117]. Such a network organization dispenses with the need for highly specific CDK–substrate interactions, making the system extremely robust. It is now clear that even CDK1, cyclin A2, and cyclin B1, all of which are essential for early mouse development, can be replaced by homologues if sufficient CDK/phosphatase activity ratio is attained. An additional level of robustness arises from the flexible rate of progression through most of the cell cycle. The division phase, mitosis, is vulnerable and needs to be achieved in a short space of time. In contrast, interphase is less risky for cells, and its length is highly variable between different cell cycle types; cells can remain virtually indefinitely in interphase. No specific CDK/cyclin complex has ever been found to be irreplaceable for passage through interphase, nor for the onset of DNA replication. The latter can be viewed less as a switch-like transition and more as the sum of activation of individual replication origins, which gathers momentum. As such, the temporal variation of CDK activity against a particular substrate of replication origins does not necessarily have drastic consequences for the cell—already activated origins can compensate for unfired ones. This difference in risk between interphase and mitosis may underlie the relatively low requirement for CDK/phosphatase activity in interphase compared to mitosis.

Viewing cell cycle control as a continuum of biological processes punctuated by the drastic reorganization at mitosis and driven by quantitative variation in the ratio of CDK to phosphatase activity recalls the binary vision of the cell cycle first exposed by Walter Flemming in 1882, in which a cell is either in interphase or mitosis. Our current molecular understanding of the complexity of the system components, interactions, and regulation sometimes obscures the underlying simplicity. A single CDK–cyclin complex may have driven the cell cycle of ancestral eukaryotes, while the current complexity of CDKs and cyclins may have arisen from marginal advantages selected over evolutionary time scales. An increase in the component number and diversity increases the dynamics and thus robustness, perhaps allowing us to adapt to changing environments, analogously to the current complexity of automobiles when compared to the 1885 prototype of Daimler and Benz.

* Answer: three: one to hold the bulb and two to turn the ladder.

## Figures and Tables

**Figure 1 cells-11-02019-f001:**
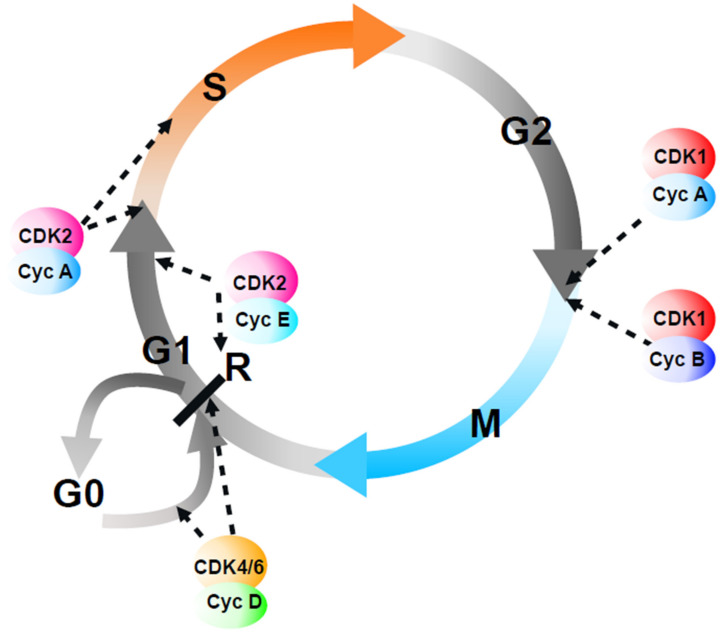
A popular representation of cell cycle control by CDKs. In this “popular” model, the vertebrate cell cycle is viewed as a unidirectional passage through four distinct phases, each one controlled by specific CDK–cyclin complexes that act sequentially (dashed arrows). At a point in mid-G1, the “restriction point” (R), if CDK2 and CDK4/6 activities are sufficient, cells continue into the next cell cycle; otherwise, they exit the cell cycle into a quiescent state (G0). As discussed in the text, this model has numerous flaws.

**Figure 2 cells-11-02019-f002:**
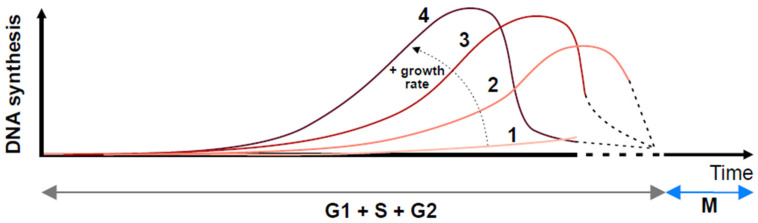
A continuum model of the cell cycle. In this model, the cells are not viewed as being at a point within, or outside, a cycle but in only one of two states, interphase (which is the sum of G1, S and G2 phases of the popular model) or mitosis. In interphase, DNA synthesis occurs at a rate that depends on the global biosynthetic capacity (growth rate). Four different rates are shown (1–4). DNA synthesis rates are not constant but increase exponentially as more replication origins are activated, and then decrease and are completed (dashed lines) prior to mitosis. In the popular model, the low DNA synthesis rates on either side of the peak, which are not detected macroscopically, correspond to G1 and G2. Curve 1 describes an extremely low biosynthetic rate; cells have not exited the cycle but synthesize DNA at such a slow rate that no macroscopic S-phase is experimentally seen. Curves 2–4 describe cells that appear, according to the popular model, to have variable G1 and G2 phase lengths.

**Figure 3 cells-11-02019-f003:**
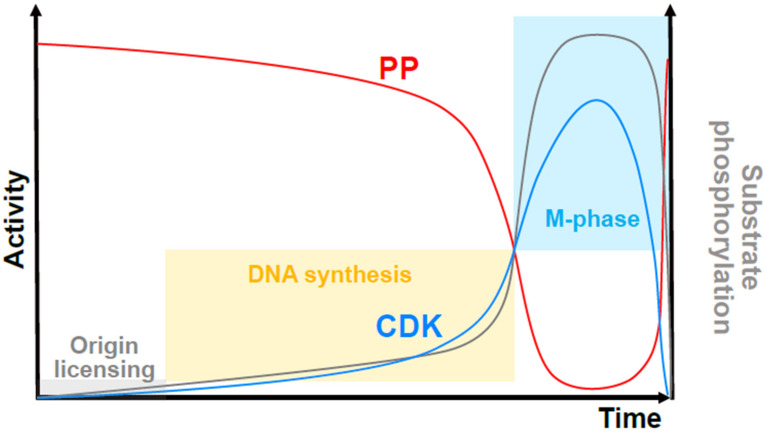
An updated quantitative model of cell cycle control by the CDK/phosphatase activity ratio. Overall CDK activity (blue curve) competes with protein phosphatase (PP) activity (red curve) to the phosphorylate substrates (grey curve shows the level of substrate phosphorylation). When substrate phosphorylation is very low (grey shaded box), DNA replication origins can be assembled (origin “licensing”). Increased phosphorylation promotes origin activation and DNA synthesis (yellow shaded box) but high levels, which are attained in a switch-like manner at a threshold of CDK to PP activity, prevent replication, and trigger mitosis (blue shaded box). The level of CDK activity required for mitotic onset is higher than that required to maintain the mitotic state.

## Data Availability

Not applicable.

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
