# Peer review of "Explaining Redundancy in CDK-Mediated Control of the Cell Cycle: Unifying the Continuum and Quantitative Models"

_cells, 2022, doi:10.3390/cells11132019_

Round 1

Reviewer 1 Report

In this review article, Fisher and Krasinska describe a cell cycle control mechanism based on CDK-cyclin driven qualitative model. Although this model has been already described, they refresh this concept with updated reports and further incorporate an idea of continuum model that the interphase progression is a continuous process leading to and performing DNA synthesis. Inherently, this is an interesting review for the readers of cell cycle and wider community.

Here are my comments.

1. From the title, “Explaining redundancy in CDK-mediated control of the cell cycle”, I wonder, readers may expect that in this review the authors focus on their idea describing why so many CDK-cyclins are involved and required for a cell to divide. Actually, how do authors think about why cells possesse so many CDK-cyclins?

2. I wonder how the high CDK activity is induced coupled to the completion of DNA replication in a quantitative model. Authors suggest phosphatase activity is one regulator. A reference reporting a high PP activity in S-phase needs to be cited. On the other hand, it can be predicted that feedback from ongoing DNA replication keeps CDK at moderate levels. Regarding this view, is there any report, maybe in fission yeast, showing how CDK activity oscillates during a cell cycle in a cell driven only by Cdc13 -Cdk1 as CDK and with or without Cdc6 expression, in the latter case, no DNA replication occurs.

3. Line 113-

“This suggests a simple continuum model, whereby growth conditions promote activity of an overall regulator, such as CDK activity, determining the rate of progression through DNA synthesis, the only abrupt transitions being the entry into, and exit from, mitosis (Figure 2).”

For readers, including me, I guess, the “continuum model” was not familiar as compared to qualitative model. The phrases relating this model appear to be presented scattered in the text. Is it possible to define and describe continuum model in more simple words.

4. For wider readers to follow this article, it is better to describe the popular model of cell cycle control” in more detail, it can be linked to section 2, or cite the Review by Editors of current special issue, Drs Uzbekov and Prigent.

A Journey through Time on the Discovery of Cell Cycle Regulation.

Uzbekov R, Prigent C. Cells. 2022 Feb 17;11(4):704. doi: 10.3390/cells11040704. PMID: 35203358 Free PMC article. Review.

5. Line 385-388;

“sequential activation of different CDK-cyclin complexes promotes successive transitions between discontinuous cell cycle phases, fails to account for most of the observations resulting from genetic analysis of CDKs and cyclins”

It is right that authors described in this phrase, but it can be said that “sequential activation of different CDK-cyclin complexes promotes successive transitions between discontinuous cell cycle phases for an eukaryotic cell in a normal genetic background”.

6. To update this review, the paper just published by Nurse’s lab can be cited.

Core control principles of the eukaryotic cell cycle.

Basu S, Greenwood J, Jones AW, Nurse P. Nature. 2022 Jun 8. doi: 10.1038/s41586-022-04798-8. Online ahead of print. PMID: 35676478

Another points:

Figure 2.  The illustration shows that the total lengths of G1+S+G2 are same irrespective of the growth rates. For a cell of lower growth rate, it will take more time to start and finish DNA replication and would have a longer G1+S+G2 interphase.

Figure 3. It is better to change the grey color of “Origin licensing” and shaded box to prevent a tangle with grey line.

Line 207, 209 and 347; “all three” “all five”

Readers would be acknowledged by mentioning D1, D2 and D3-cyclins for three D-type cyclins, and E1 and E2 for E-type cyclins.

Line 211.  I cannot understand which “the latter” correspond to?

Line 340.  “by inhibiting CDC25”. Is “inhibiting” correct?

Line 423. Does “simplicity” get to the idea that primordial eukaryotic cell division is driven by a single CDK-cyclin?

Reviewer 2 Report

In this work Fisher and Krasinska review published data to challenge the classical model of cell cell division and support a model in which the cell cycle is not a cycle per se but rather a rheostat system comprising interphase and mitosis.

In the classical model of cell cycle regulation, the activity and expression of specific CDK/cyclin couples is timely regulated and responsible for the progression through separated G1/G0, S, G2 and M phases. However, in many organisms and cell types these phases overlap to a large extend, which is enough to challenge the unified classical model. Therefore, the authors suggest that G1, S and G2 belong to a single “interphase” state, where G0 would simply be an elongation of interphase  and not a specific phase (if conditions do not allow for mitosis entry). Furthermore, in many biological systems there is a high level of redundancy between CDKs. Therefore, instead of involving a timely upregulation of specific CDKs, mitosis entry could be under the control of an equilibrium between CDKs and phosphatases activities. In this model, CDK activity and substrate phosphorylation increases until reaching a certain threshold when mitosis entry is inevitable. In parallel, phosphatase activity decreases. This model thus recapitulates the biphasic vision that Flemming had back in 1882 – the cycle is closed.

This review article is very well written, interesting and pleasant to read. It provides an alternative to the popular model for cell division regulation which is supported by evidences coming from multiple experiments.

I have no major comments.

Minor comments:

1)    “Cell cycle CDKs” have been shown to also regulate the basal transcription machinery during cell cycle. If the authors find it relevant, could they discuss whether a global increase in CDK activity towards mitosis is compatible with transcriptional regulation during interphase in their model?

2)    The figures could benefit from a better design. For example, I could not read the text in the graphical abstract due to low resolution and inadequate color matching. 
